# Invisible Barriers: Institutional Discrimination Against Asylum-Seeking Women in Portugal

**DOI:** 10.3390/healthcare13222967

**Published:** 2025-11-19

**Authors:** Gabriela Mesquita Borges

**Affiliations:** 1Center of Legal, Economic and Environmental Studies (CEJEIA), Lusíada University, R. de Moçambique 21 e 71, 4100-348 Porto, Portugal; gabrielamborges@por.ulusiada.pt; 2Communication and Society Research Centre (CECS), University of Minho, 4710-057 Braga, Portugal

**Keywords:** asylum seekers, refugee women, discrimination, integration, gender-based violence

## Abstract

**Introduction:** Building a life in an asylum country poses specific challenges for women, who often face compounded barriers in healthcare, language acquisition, economic independence, childcare, education, cultural adaptation, and legal processes. This study examines the experiences of asylum-seeking women in Portugal, focusing on discrimination perpetrated by professionals within reception and integration institutions. **Methods:** Drawing on 24 semi-structured interviews with women from the Middle East (*n* = 14) and Africa (*n* = 10), this research adopts a criminological and gender lens and employs a narrative paradigm informed by constructivist Grounded Theory and an abductive approach. **Results:** The analysis reveals that institutional discrimination, manifested through neglect, hostility, and cultural insensitivity, reinforces feelings of abandonment and fear, obstructs integration, and perpetuates cycles of marginalization and vulnerability. These dynamics are intensified by gender-based and structural violence embedded in asylum procedures and professional practices. The findings highlight the emotional and relational dimensions of institutional encounters, showing how empathy, trust, and intercultural awareness among professionals are crucial for effective inclusion. **Conclusions:** This study concludes that addressing institutional discrimination requires systemic change, professional training in gender-sensitive and intercultural competencies, and the promotion of equitable, inclusive, and human rights-based reception practices in Portugal.

## 1. Introduction

The 1951 Convention Relating to the Status of Refugees represents a comprehensive milestone in the codification of refugee rights at the global level. It establishes a clear definition of a refugee as an individual or group compelled to leave their country due to various causes, such as violence, torture, or external pressures arising from armed conflicts, as well as political, religious, or social reasons.

Under international law, the distinction between refugees and asylum seekers is fundamental: while the 1951 Convention and its 1967 Protocol outline the criteria for refugee status, an asylum seeker is a person who has applied for international protection and whose request has not yet been decided.

In the Portuguese legal framework, this concept is defined by Law No. 26/2014 of 5 May (Asylum Law), which transposes the European Union’s Asylum Procedures Directive (2013/32/EU). According to Article 2(1)(d), an asylum seeker (requerente de asilo) is an individual who has submitted an application for international protection and awaits a final decision. Thus, in Portugal, the legal definition aligns with the broader EU and UNHCR framework while reflecting the country’s specific administrative procedures.

It is important to note that, although all refugees begin their journey as asylum seekers, not all will ultimately be recognized as such, as this is a process subject to rigorous scrutiny.

At present, it is widely acknowledged that asylum seekers and refugees deserve a legal framework ensuring the protection and realization of their rights, as established by international law. The granting of asylum is intrinsically linked to fundamental historical norms, such as the 1948 Universal Declaration of Human Rights [1]. The 1951 Convention, together with its 1967 Protocol and the United Nations High Commissioner for Refugees (UNHCR), sets out specific rights, freedoms, and minimum standards of treatment for foreigners in host countries, encompassing areas such as freedom of movement, religion, association, welfare, employment, administrative assistance, and property rights [2].

States that are parties to the 1951 Convention and its 1967 Protocol are required to cooperate with UNHCR, fulfil their assigned obligations, and respect refugee status. However, they are not legally bound to grant asylum and are prohibited from returning individuals to territories where their life, safety, or well-being would be at risk [3].

Furthermore, the principle of non-discrimination, rooted in Article 2 of the Universal Declaration of Human Rights and reflected in Article 3 of the 1951 Convention, guarantees equal treatment for asylum seekers and refugees without distinction of race, religion, country of origin, sex, age, disability, or other prohibited grounds. This principle is reinforced by the 1966 International Covenant on Civil and Political Rights, which emphasizes the importance of preventing discrimination to safeguard a wide range of fundamental rights [4].

Within this context, this article aims to examine discrimination against female asylum seekers in Portugal from a gender perspective, drawing on Oberman’s conceptualization of discrimination as treating individuals worse than others because of their membership in a socially salient group [5]. Socially salient groups are those whose membership is fundamental to the structure of social interactions across various social contexts [5]. This article seeks to explore discriminatory practices within the Portuguese asylum regime itself, particularly through the actions of professionals working in institutions responsible for the reception and integration of asylum seekers. Based on data collected from 24 interviews with female asylum seekers within the Portuguese asylum system, this article provides critical insights into the challenges faced by asylum seekers in Portugal and the role of institutions and professionals in either perpetuating or mitigating discrimination.

Nevertheless, it is important to acknowledge that the complexity of this issue presents a significant challenge, requiring a holistic and collaborative approach to effectively promote the inclusion and protection of asylum seekers and refugees within Portuguese society.

### Background

Beginning in 2014, Europe faced a refugee crisis that triggered intense debates on migration, revealing the tensions between humanitarian concerns and the protection of the European Union’s borders [6,7]. Political commentators have often portrayed irregular migration—and, at times, even the arrival of asylum seekers and refugees—as a threat, blurring the distinction between people migrating without authorization and those seeking international protection [8,9,10,11,12,13].

Several scholars have explored the relevance of Becker’s work on outsiders and its application of labelling theory to asylum seekers and refugees [14,15,16,17,18]. According to Becker’s premises, deviance and labels are socially constructed through political processes in which certain groups impose their perspectives as more legitimate. Becker’s interactionist theory [14,15] shifts criminology’s focus from crime itself to deviance as a social relationship, in which deviance results from the labelling of acts as deviant, shaping the self-perception and behaviour of those labelled as outsiders.

Cohen’s theory of moral panic involves labelling certain conditions or groups as threats, disproportionately amplifying perceived dangers [9]. This phenomenon occurs when a condition, event, or group is exaggeratedly perceived as a threat to societal values and interests, resulting in an excessive and disproportionate response [9]. Globalization extends local moral panics, while concerns about community control [19] highlight the blurred boundaries between deviants and non-deviants, affecting asylum seekers through detention and restrictive measures.

Moreover, there is often no clear distinction between the categories of deviant and nondeviant individuals, and within groups considered deviant there exists significant diversity in characteristics and behaviours [19]. More recently, Aas has examined social anxieties surrounding immigrants and border security, leading to exclusionary asylum systems and introducing the concept of the crimmigrant other [20].

The penal power exercised against crimmigrant others is portrayed as a moral force, transforming issues of global privilege into matters of morality and thereby contributing to the spread of moral panic. Aas underscores the importance of understanding “immigrant criminality” as a means of assessing governments’ and societies’ capacity for empathy toward outsiders, challenging existing borders and prompting a comprehensive analysis of crimmigration practices [20].

Martin notes that moral panic can evolve from a temporary state into a relatively permanent condition, as exemplified by the “war on terror,” in which Muslim refugees have been depicted as a transnational threat. The widespread negative response to refugees and their demands for international protection exemplifies a classic moral panic [11,21].

In the context of receiving asylum seekers and refugees, professionals working in institutions responsible for their reception and integration play a crucial role in upholding the principle of nondiscrimination. Their work involves not only providing means of subsistence but also addressing the traumatic dimensions that often characterize the experiences of these individuals. This responsibility extends beyond material provision, encompassing the creation of a safe and trustworthy environment for those seeking protection [22,23,24].

The successful integration of asylum seekers and refugees extends beyond bureaucracy and logistics; it also involves the establishment of healthy and empathetic relationships between these vulnerable individuals and the professionals working within reception institutions. Recognizing the importance of such relationships is essential, as they play a crucial role in facilitating a smooth and effective transition for asylum seekers.

Despite the challenges involved, these professionals are expected to fulfil their duties with empathy and integrity, thereby contributing to the positive integration of asylum seekers and refugees into society [25,26]. However, the inherent complexity of this task poses a significant challenge, as will be demonstrated in this paper.

## 2. Methodology

### 2.1. Sample

The sample intentionally included women of diverse nationalities, ages, family contexts, and lengths of stay in Portugal, reflecting heterogeneous experiences of migration and integration. Although psychological or criminological characteristics were not used as selection criteria due to ethical and methodological considerations, the sample’s variation in socio-demographic and institutional conditions provided a broad and rich range of perspectives on discrimination and adaptation processes.

This study comprised 24 refugee women, of whom 11 were from Syria, 3 from Iraq, 3 from Angola, 2 from the Democratic Republic of the Congo, and one each from Togo, Sudan, Somalia, The Gambia, and Morocco. The participants’ ages ranged from 21 to 48 years. Regarding marital status, 66.66% were married, 20.83% were single, and 12.5% were divorced. The number of children ranged from none to six. Approximately 79.16% of the women had limited formal education, having mostly completed only lower secondary education. However, five of them held a bachelor’s degree.

Upon arrival in Portugal, 70.83% of the participants were classified as asylum seekers under the EU resettlement and relocation programmes, while 29.16% were spontaneous asylum seekers. The average length of stay in Portugal ranged between two and four years. One woman, who arrived at the end of 2014, had the longest period of residence and was the only spontaneous asylum seeker to have obtained refugee status at the time of data collection.

### 2.2. Data Collection and Analysis

Refugee women residing in Portugal were selected for this study through a purposive sampling procedure [27,28], with the assistance of professionals from relevant institutions, including social workers, case managers, and NGO staff members working in reception and integration centres under the supervision of the Portuguese Refugee Council (Conselho Português para os Refugiados) and the High Commission for Migration (Alto Comissariado para as Migrações). These professionals facilitated contact with potential participants by informing them about this research and assisting in the recruitment process while ensuring confidentiality and voluntary participation.

The selection criteria included adult age (18 years and beyond), asylum protection status, arrival during the European refugee crisis (from 2014 to 2019), and either proficiency in Portuguese or English or consent to have a translator present. No additional exclusion criteria were applied, as this study sought to include all adult female asylum seekers meeting these basic conditions. However, individuals who did not provide informed consent or whose asylum process had already been concluded with long-term refugee status prior to 2014 were not included.

Data collection consisted of semi-structured interviews conducted using a narrative criminology approach [29,30,31]. The data collection period extended from January 2020 to April 2021. Thirteen interviews were conducted with the assistance of two translators. Due to the COVID-19 pandemic, most interviews were conducted online, with only five face-to-face interviews being possible under the containment measures in place. The interviews focused on the women’s experiences during the asylum application process, and participants expressed a strong willingness to share their stories, believing that their testimonies could help others in similar situations.

The researcher personally transcribed all interviews, while translators provided oral translations from Arabic and French into English prior to transcription. Field notes capturing nonverbal cues and environmental details were incorporated into the transcripts. This study adopted a constructivist grounded theory approach, emphasizing the participants’ construction of meaning [32,33], complemented by an abductive approach to uncover emerging concepts.

The data were analyzed thematically through an inductive process led by the author. The coding process involved identifying recurring themes, selecting subthemes, and employing focused colour-coded categorization [33]. Categories were developed through the simultaneous generation and analysis of data, including intermediate coding. Although the coding was not independently replicated, the analytic framework was refined through iterative comparisons and peer debriefing with colleagues specializing in migration and refugee studies. This approach enhanced interpretive consistency and transparency while remaining faithful to the principles of reflexive qualitative analysis, which prioritize meaning and context over statistical reliability. The interpretive procedure’s main strength lies in its capacity to produce contextually rich, theoretically grounded insights into participants’ lived experiences. A potential limitation is the influence of the researcher’s interpretive lens, mitigated through reflexivity and the inclusion of direct quotations to ensure that participants’ voices remain central.

To enhance transparency in how grounded theory was applied, the following examples illustrate the analytical reasoning process that guided this study. They demonstrate how raw data were transformed into increasingly abstract concepts through successive phases of coding—initial, focused, and theoretical—while maintaining close connection to participants’ lived experiences. By providing these examples, this study makes explicit how the inductive logic of grounded theory, supported by constant comparison and memoing, informed the generation of categories and theoretical insights from empirical observations.

During an interview with a refugee woman who was living in a Portuguese reception center, a professional working in that reception center entered the private room where the interview was being conducted, to check whether the internet connection was working properly. In another case, the refugee woman suddenly ended the interview because a professional from the institution in charge of her asylum process entered the shared apartment she was living in, and she was afraid to continue to speak with the interviewer because she did not want the professional to hear complaints in relation to the institution’s practices. These two incidents were translated into a specific initial code, namely “the reaction of women to the interruptions of the interviews by professionals”; then, through focused coding, it was considered part of a broader category, “the type of relationships formed between refugee women and the professionals from the institutions.” This broader category was then used as a subcategory—“professionals’ behavior”—included in a core category, “integration practices.”

This example exemplifies the inductive and iterative logic of grounded theory. Through constant comparison, an incident initially coded as “the reaction of women to the interruptions of the interviews by professionals” was progressively abstracted into the subcategory “professionals’ behavior,” which was later integrated into the core category “integration practices.” This process illustrates how data collection and analysis occurred simultaneously, with each phase of coding deepening theoretical understanding. By making this progression explicit, this study highlights how grounded theory methodology informed not only data organization but also the substantive development of the theoretical constructs discussed in the Results section.

To manage the large volume of collected data, the qualitative data analysis software QDA Miner (version 6.0) was used.

In line with Small and Calarco’s (2022) [34] proposal for enhancing the validity of qualitative research, this study was guided by three key criteria: empirical adequacy, theoretical contribution, and reflexive transparency. Empirical adequacy was ensured through the careful transcription of all interviews, the inclusion of field notes, and the constant comparative analysis that grounded each conceptual category in participants’ lived experiences. Theoretical contribution was achieved through the development of core categories—such as “integration practices” and “professionals’ behavior”—that extend existing understandings of institutional discrimination in asylum contexts. Reflexive transparency was maintained through continuous memo writing, ethical reflection, and peer debriefing, which helped the researcher to identify and mitigate interpretive bias. Together, these principles provided an external framework for assessing the rigor and credibility of the analytic process. The following section further develops this dimension, offering a detailed account of the researcher’s reflexive stance and ethical considerations that guided this study.

### 2.3. Reflexivity and the Ethical Framework

This study was approved by the Research Ethics Committee of the Faculty of Law at the University of Porto (Portugal). However, ensuring ethical research extends beyond the mere approval of an ethics committee. By adopting a reflexive stance [35], the researcher was able to fully consider the guiding principles that underpin the integrity of the research process [36]. Ethical reflection involves recognizing micro-level dimensions, managing tensions, and protecting participants [9].

The researcher employed reflexivity to explore a range of issues, including the use of translators during interviews, the emotional impact of this research on the researcher, the relationship between the researcher and participants, data collection during a global lockdown/pandemic [9], and the influence of gatekeepers on participant recruitment [37].

Conducting research with asylum-seeking women also required a high degree of ethical awareness, empathy, and reflexivity, given the participants’ vulnerability and the sensitive nature of their experiences. Careful consideration was given to obtaining informed consent, ensuring anonymity, and protecting participants from any potential harm arising from the disclosure of traumatic experiences.

Participants were informed about the objectives and procedures of this study and provided their informed consent. Audio recordings were made, and most interviews were conducted online via Zoom. Pre-interview sessions addressed technical aspects. Data were securely stored and deleted following transcription. Participants were assured of their right to withdraw from this study at any time should they wish to do so, and were informed that any images or audio recordings would be immediately erased upon withdrawal.

Most of the interviews were conducted online due to logistical constraints and, in some cases, health and safety considerations. While remote interviewing can alter the dynamics of empathy and non-verbal communication [34], this approach also allowed participants to engage from familiar and comfortable environments, often leading to candid and reflective conversations. Throughout the process, particular attention was given to maintaining an empathetic tone, building trust, and ensuring that participants felt listened to and respected. As a result, the online format did not appear to diminish the depth or emotional quality of the narratives shared.

Furthermore, participants were encouraged to remain alone during the initial interviews; however, this decision was ultimately left to them to avoid feelings of coercion and to prevent revictimization or additional trauma [38]. Safety measures were implemented to protect against potential third-party risks. These included conducting interviews via real-time video calls and providing participants with a “safety word” they could use to alert the researcher if they needed to stop the interview or felt threatened by the unexpected presence of others.

It is important to note that no major incidents occurred during the interviews. The only minor incident was an unexpected interruption when a social worker from the reception institution responsible for one participant’s asylum case entered the apartment where the interview was taking place without prior notice. The participant later explained—via a message to the researcher—that she had abruptly ended the interview because she did not want the social worker to overhear her complaints about the reception institution, fearing possible retaliation.

## 3. Results and Discussion

The findings of this study reveal that when the professionals working in reception and integration institutions—such as social workers, case managers, and NGO staff—neglect or inadequately manage their relationships with asylum seekers, a range of challenges may arise, from simple language barriers to complex legal issues [39,40]. Furthermore, a lack of mutual understanding and connection can lead to broader problems such as unemployment and social exclusion [41].

This relational dimension assumes even greater significance in the case of female asylum seekers. The United Nations Division for the Advancement of Women [42] highlights the particular vulnerability of migrant women to deprivation, suffering, discrimination, and abuse. The concept of intersectionality underscores the dual vulnerability of women seeking asylum to violence, considering both their legal status and gender. These women often face specific and additional challenges, including trauma resulting from gender-based violence, sexual abuse, or exploitation [43]. For many, trust and empathy from reception professionals are essential in overcoming such traumatic experiences and beginning to rebuild their lives in a new environment. Moreover, female asylum seekers may have distinct health needs, including reproductive healthcare, psychological support, and protection from gender-based violence.

The study discussed in this article was based on the fundamental assumption that professionals working in reception institutions must be able to establish relationships of empathy, trust, and support with female asylum seekers in order to effectively fulfil their responsibilities regarding reception and integration. However, most participants described their relationships with institutional professionals as unsatisfactory or unpleasant, frequently using terms such as abandonment, hostility, and neglect to characterize the support they received.

For instance, Madalena (Angola) stated:

“When I arrived in this country, I came with the hope of restarting my life, but the way I was treated by the institution that received me made me feel completely forgotten.”

This quote vividly illustrates the disconnect between the initial expectations of support and the lived reality experienced by Madalena and other women in similar situations. The experience shared by Ama (Togo) further highlights the hostility encountered:

“At the first institution where I stayed, there was a doctor who was really unkind. She didn’t give us proper attention, didn’t help us at all, and showed very little empathy for our situation.”

Additionally, Rana (Syria) reported that the professionals responsible for her reception only visited her at the apartment where she had been housed:

“Eight months after my arrival in this country, just to see the conditions I was living in. That makes no sense to me. It’s true they found me a place to live, but after that, they stopped caring. They didn’t help me with anything.”

These testimonies collectively reflect a broader pattern of emotional and institutional disconnection between asylum-seeking women and the professionals tasked with their care—an issue that raises important questions about the structural and interpersonal dimensions of Portugal’s asylum system.

By generating feelings of abandonment, hostility, and neglect, professionals fail to provide asylum seekers with a sense of protection. Such neglect can lead to various adverse outcomes, including frustration, trauma, and mental health issues—particularly post-traumatic stress disorder.

(PTSD) and depression [44]. Wenzel and Drozdek [26] emphasize that, during the integration process, the fear of insecurity often replaces the fear of annihilation triggered by the displacement journey. Even after the immediate risks have been eliminated in resettlement countries, the impact of traumatic and violent experiences persists. In addition, feelings of emptiness and disorientation are common among displaced persons, making smooth integration processes more difficult [23], as also observed in the present study.

Regarding the support effectively provided, the participants shared several instances where assistance proved inadequate or insufficient. Hanan (Syria) recounted a particularly distressing situation:

“One night, my daughter developed a very high fever. I contacted a woman who worked at the institution that had taken us in, and she called an ambulance to my apartment. However, when we arrived at the hospital, I realized she wasn’t there waiting for us. I had to explain everything to the doctors as best I could, in a language I still couldn’t speak properly. I felt completely helpless and unsafe, not knowing what was wrong with my daughter or what the doctors were telling me. And later, when my daughter was discharged, I felt completely lost because I had no idea where in the city I was, nor how to get back to my apartment.”

Hanan’s (Syria) account underscores the lack of adequate support in the face of a medical emergency and illustrates the obstacles faced by individuals in vulnerable situations when the assistance provided fails to meet their urgent needs. Similarly, Jassim (Syria) shared an experience related to healthcare:

“Once, I was walking down the street with one of the social workers responsible for my case, and I suddenly fainted. When I regained consciousness, she didn’t seem particularly concerned about me. The fainting spells became more frequent. I reached a point where I couldn’t leave the house alone. I repeatedly asked this woman for help, but she did nothing to assist me—she only promised to schedule a doctor’s appointment, which she never did. I eventually went to the doctor on my own.”

The experiences shared by participants reveal authoritarian and aggressive behaviour on the part of some professionals, provoking negative emotions, with fear being the most common. As Alice (Angola) stated:

“I vividly remember situations where professionals displayed a hostile attitude, leaving me with an overwhelming sense of fear and vulnerability. Unfortunately, these were not isolated cases—it seems to be a reality shared by many of us.”

This fear was even evident during one of the interviews, when a participant abruptly ended a video call after being unexpectedly interrupted by a professional from the institution responsible for her reception. As described earlier, this incident illustrates how the perceived risk of surveillance and retaliation can silence asylum-seeking women, reinforcing an atmosphere of mistrust and fear.

Fear is an intense emotion that, when misused, can be profoundly disabling and harmful [45]. Within the Portuguese asylum system, there is evidence of a “culture of fear” [46], in which professionals resort to fear as a means of manipulating and controlling asylum seekers. This use of fear severely undermines the establishment of trust between professionals and individuals seeking asylum. Moreover, this dynamic reflects Foucault’s [47] concept of the “docilization of bodies,” in which power is exercised to shape individuals according to the expectations of those in authority. A concrete example of this issue was described by Ama (Togo), who faced a particularly distressing situation:

“I spent two nights away from the centre taking care of a sick friend. When I returned, I was told that I could no longer stay there. The manager started shouting at me, threatening to call security if I didn’t leave immediately. She insisted that I had to follow her rules and that I was not allowed to act on my own will. Even after I explained the situation, she kept yelling, saying that I had disobeyed her orders and therefore had to go. When I said I had nowhere to go, she told me that wasn’t her problem.”

Several authors argue that effective management of cultural diversity brings more benefits than challenges to host societies [48,49,50,51]. However, as revealed in other studies [52,53], the narratives of several participants in this research also highlighted the presence of cultural clashes between them and the professionals. These tensions manifested in relation to food:

“Sometimes, I go to bed without eating, or I eat and then vomit because I’m not used to Portuguese dishes and spices. I’ve asked several times to be allowed to cook my own food, but in this institution, meals are handled by the canteen, and my request was denied.”(Promesse, Congo)

—in relation to clothing:

“I was advised that when I went looking for a job, I should dress in a way more compatible with Portuguese culture.”(Fatima, Iraq)

—and even concerning motherhood, as demonstrated in Shamshi’s (Somalia) account:

“One of the staff members at the institution became quite upset when I told her I was pregnant with my fifth child. She even told me that, given my circumstances, I should have been more careful and avoided becoming pregnant. From what she said, a child is seen as an expense. However, in my culture, a child is always a blessing, because there is nothing more valuable than family.”

These examples illustrate the clash between institutional expectations and the cultural identities of asylum-seeking women. The imposition of dominant cultural norms—whether through explicit control, condescension, or moral judgment—creates environments of subordination rather than inclusion. This not only reproduces asymmetrical power relations but also reinforces the marginalization and vulnerability of women who are already navigating the hardships of displacement and resettlement.

For most participants, the shortcomings identified in the support provided by institutional professionals stem primarily from a deep mistrust regarding the true motives behind asylum claims, as explained by Hilal (Syria):

“The professionals who work in reception institutions have a negative opinion of us. They often confuse us with economic migrants and think we are here to steal their jobs—or worse, to harm them. In any case, I believe my legal status should be completely irrelevant. They [referring to institutional professionals] should do their work properly, and their job is to provide assistance, not to judge the reasons that led me to flee my country. The truth is, I felt discriminated against by some people.”

The emphasis on social integration is crucial to promoting cohesion and solidarity among migrants. However, the “us versus them” stance often adopted by European states contributes to segregation and hinders the effective integration of asylum seekers and refugees into society, frequently leaving them dependent on themselves for survival. This scenario raises concerns about the effectiveness of current integration policies and highlights the urgent need to address the structural barriers preventing the full and equal inclusion of asylum seekers and refugees within host communities [48].

Durkheim’s [54] structural functionalism conceptualizes society as a living organism, emphasizing the vital functions performed by its intrinsic structures. When applied to the issue of asylum seekers, despite the potential benefits they may bring to European societies, they tend to be perceived as a problem rather than a solution. The driving force behind this perspective lies in transforming global concerns into moral issues and maintaining moral order through the social stigmatization of the “crimmigrant” [20].

Bauman [8] described the challenges arising from the migration crisis of the mid-2010s as extremely complex and controversial, in which the moral imperative is directly confronted by the fear of the “great unknown,” personified by the “masses of strangers” at Europe’s borders. According to Robertshaw, Dhesi, and Jones [55], professionals working directly with asylum seekers in any institution have the responsibility to recognize that these individuals, who have experienced forced migration, likely possess highly diverse needs and experiences. They must also acquire new skills and adapt to the emerging challenges posed by the most recent refugee crisis [56].

However, as evidenced in this article, the discriminatory attitudes reported by participants demonstrate the persistence of prejudice and significant challenges in the treatment of asylum seekers. This situation is particularly concerning in the case of women, as highlighted by Canning’s [43] concept of the “continuum of intersectional violence,” which posits that asylum-seeking and refugee women experience multiple forms of violence throughout their lives and that not all women experience violence in the same way.

Borges [57,58,59] also draws attention to the multiple forms of violence faced by refugee women, including gender-based, structural, and symbolic factors. Based on findings derived from the same dataset used in this article, Borges [57,58,59] proposed the conceptualization of “violence against refugee women” as encompassing a continuum of various forms of violence—gender-based, structural, and symbolic—that result in physical, psychological, sexual, and emotional harm. These forms of violence persist across different stages and locations: in the country of origin, during displacement, and in the host country [57,58,59].

In the context of the host country, such forms of violence perpetuate harmful discourses among institutional professionals, who inevitably fail to meet the needs of asylum-seeking and refugee women. This failure contributes to their exclusion, segregation, and marginalization [60,61,62]. Structural and symbolic factors further amplify the vulnerability of women who already occupy precarious positions, rendering them more susceptible to gender-based violence and social invisibility within asylum and integration systems.

## 4. Conclusions

The findings reveal that asylum-seeking women’s interactions with host society institutions are complex and multidimensional. Beyond the practical difficulties of accessing services, these encounters encompass emotional, relational, and gendered dimensions that profoundly shape their sense of belonging and trust. Institutional structures often determine the degree of support or exclusion experienced, while the attitudes and behaviours of professionals mediate how asylum seekers perceive fairness, empathy, and respect. These interrelated dimensions—structural, interpersonal, and emotional—collectively define how processes of integration are lived and understood by participants.

The challenges faced by asylum-seeking women in Portugal reflect the intricate interplay between prevailing social attitudes and the actions of the institutions and professionals responsible for their reception and integration. The persistence of negative or stereotypical portrayals of asylum seekers contributes to a climate of “moral panic,” which can intensify discrimination and obstruct inclusion. Within this context, the role of professionals is pivotal: their sensitivity, training, and everyday practices can either mitigate or exacerbate the vulnerabilities experienced by asylum seekers.

These findings resonate with previous research highlighting how gender, power, and institutional culture intersect to shape asylum experiences [38,39,45]. As a qualitative and interpretive study based on a limited sample, the results should be understood as context-specific and exploratory rather than statistically generalizable.

As with all qualitative research, the findings presented here are interpretive constructions rather than objective representations of reality. The relatively small and heterogeneous sample, although intentionally selected to capture a range of experiences, limits the transferability of the results to other contexts. The analysis was also influenced by the researcher’s interpretive lens, cultural background, and professional experience. To mitigate this potential bias, this study adopted a reflexive stance throughout all stages of this research, supported by continuous memo writing, peer debriefing with colleagues specializing in refugee studies, and the inclusion of verbatim excerpts to ensure that participants’ voices remained central. Moreover, the predominance of online interviews—while ensuring safety and accessibility during the COVID-19 pandemic—may have restricted the observation of certain non-verbal cues that often enrich in-person interactions. Despite these constraints, the combination of reflexive analysis, methodological transparency, and adherence to recognized criteria for qualitative validity (Small & Calarco, 2022) reinforces the trustworthiness of this study’s findings [34].

The strength of this research lies in its depth of analysis and its capacity to capture the lived experiences and perspectives of women navigating the asylum system. Nonetheless, future studies combining qualitative and quantitative approaches, or comparing different institutional settings, could provide further insight into the mechanisms underlying discrimination and integration outcomes.

Reducing discrimination and fostering meaningful integration require that professionals receive adequate training and awareness-raising on the specific needs of asylum seekers, with particular attention to the compounded barriers faced by women. The State should take responsibility for ensuring continuous education and professional development, alongside implementing measures that counter prejudice and promote an inclusive environment. While training is essential, education alone cannot fully eliminate discriminatory attitudes or practices. Broader structural, cultural, and institutional factors also shape professionals’ responses to asylum seekers. In Portugal, some introductory training initiatives have been offered by governmental bodies such as the High Commission for Migration (ACM) and the Portuguese Refugee Council (CPR), focusing mainly on intercultural mediation and integration support. However, the availability, consistency, and measurable impact of such training remain limited and have not yet been systematically assessed. Future research could therefore examine these initiatives comparatively, exploring how institutional environments, policy frameworks, and professional cultures influence the effectiveness of integration practices.

Ultimately, this study contributes to understanding how relationships grounded in trust, respect, and mutual understanding between asylum seekers and professionals can be developed and sustained. By highlighting the mechanisms through which institutional interactions shape women’s experiences of inclusion and exclusion, this research offers an interpretive framework rather than a definitive model. Such relationships not only enhance the effectiveness of integration processes but also contribute to the development of a more equitable and compassionate social order.

## Data Availability

The data that support the findings of this study are openly available in “Open Repository of the University of Porto” (Portugal), at https://repositorio-aberto.up.pt/handle/10216/148354 (accessed on 16 November 2025).

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
