# Peer review of "Invisible Barriers: Institutional Discrimination Against Asylum-Seeking Women in Portugal"

_healthcare, 2025, doi:10.3390/healthcare13222967_

Round 1
Reviewer 1 Report
Comments and Suggestions for Authors
The paper refers to a very important topic, institutional exclusionary practices affecting female refugees and asylum seekers. Choosing a single country and providing this local perspective, against broader European/global perspective of human rights and emphasising the role of intersectionalitymakes the research more interesteing. However, in the reviewer’s opinion, the paper should be improved.
The main concern is about the references. Having double checked, even triple in some cases, the reviewer cannot see quite many of the references mentioned in the text, in the “references” section. This refers to:
Goodwin-Gill & McAdam, 2007 (line 48)
Goodwin-Gill, 2011 (line 60)
Oberman, 2020 (lines 62 and 66)
Cohen, 2002 (line 83) (there are three other references by Cohen, but the one dated 2002 is missing)
Souter, 2011 (line 84)
Becker, 1971 (line 90) (there is “another” Becker, the year is different)
Wenzel&Droztek, 2019 (line 125 and 255-256)
Patton, 2015 (line 145)
Ritchie et al., 2014 (line 145)
Cowburn et al., 2017 (line 173)
Surmiak, 2018 (line 192)
Freedman, 2019 (line 211)
United Nations Division…, 2003 (line 216)
Matlin et al., 2018 (line 299)
A typo: Glasser, 201 vs Glassner, 2010 (lines 300 vs 481)
Other remarques:
The second sentence (lines 33-37): why UK is given as an example of a single country legal perspective when defining “asylum seeker” (AS)? This is not clear, moreover this information is given between the paragraphs relating to international law (1951 Convention and Protocol, 1948 Declaration etc). In the reviewer’s opinion, it could be even more clear to start with this international legal framework, then to give the single country approach example, although it is still not clear why UK. As the paper is about Portugal, what about the definition of AS in Portugal?
The aim, line 61: it could be specified that you examine discrimination against female AS in Portugal; although the Author declares they examine it from the gender perspective anyway, in the reviewer’s opinion, if would be better to use the word “woman/women/female” for the first time earlier than in the sample description, as it is also important for describing the goal(s) of the paper.
Regarding data collection and analysis, the Author mention “professionals from relevant institutions”, with no details. It would be interesting to know what kind of institutions or professionals (only social workers?) were involved (line 146). Just what kond of insitutions are they, if possible.
Data collection period extended from January 2020 to April 2021. However, the Author declares that one of the selection criteria was arrival during the crisis “from 2014 to 2022” (lines 147 vs 152).
The selection criteria included age: what age? Not specified (line 146). (Years of arrival are specified)
What about exclusion criteria? Any?
The paragraph starting with the line 215 could be placed in the background section, maybe not necessarily, but it seems this is not a part of the results of this specific research. Or some part of this paragraph could be moved to the background, as it could help to explain why the topic of the paper is so important.
Lines 289-292 seem to be the same situation as described in the paragraph 197-203. Understanding the need of mentioning this case in both places, the reviewer is not sure if the whole situation has to be described twice.
Additionally, line 197, it is not clear what “incident” means if the case of interruption happened. Maybe it should be said that no “major” incidents occurred. Just for rethinking.
Mentioning challenges arising from the current “migration crisis” (line 361-362), quoting Bauman, is it “current” from Bauman’s or the Author’s perspective? It means, 2016 or 2025 perspective?
Line 391: “Parte superior de formulario”. Should it appear here?
As for the Conclusions, the role of training is emphasized, however it is interesting if only/mainly education/training can improve the situation. This could be probably a large topic for another research (assessment of effectiveness of trainings, but also other factors influencing the professionals behaviour, attitudes leading to discrimination). Education and training is recommended, however there is no information included, if any training for those professionals is offered in Portugal.
References:
Bekyol & Bendel, 2016 – seems not to be included in the text. Please double check.
Author Response
Response to reviewer:
- The main concern is about the references. Having double checked, even triple in some cases, the reviewer cannot see quite many of the references mentioned in the text, in the “references” section. This refers to:
- Goodwin-Gill & McAdam, 2007 (line 48)
- Goodwin-Gill, 2011 (line 60)
- Oberman, 2020 (lines 62 and 66)
- Cohen, 2002 (line 83) (there are three other references by Cohen, but the one dated 2002 is missing)
- Souter, 2011 (line 84)
- Becker, 1971 (line 90) (there is “another” Becker, the year is different)
- Wenzel&Droztek, 2019 (line 125 and 255-256)
- Patton, 2015 (line 145)
- Ritchie et al., 2014 (line 145)
- Cowburn et al., 2017 (line 173)
- Surmiak, 2018 (line 192)
- Freedman, 2019 (line 211)
- United Nations Division…, 2003 (line 216)
- Matlin et al., 2018 (line 299)
- A typo: Glasser, 201 vs Glassner, 2010 (lines 300 vs 481)
Author Response: Thank you very much for this careful and detailed observation. Following the reviewer’s comment, the entire reference list and all in-text citations were systematically reviewed and cross-checked to ensure full consistency and accuracy. Several missing references have now been added to the References section, including works by Goodwin-Gill & McAdam (2007); Goodwin-Gill (2011); Oberman (2020); Cohen (2002); Souter (2011); Becker (1971); Wenzel & Drozdek (2018); Patton (2015); Ritchie et al. (2014); Cowburn et al. (2017); Surmiak (2018); United Nations Division for the Advancement of Women (2003); and Matlin et al. (2018). Additionally, minor typographical inconsistencies were corrected (e.g., Glassner instead of Glasser), and the reference for Wenzel & Drozdek was updated to its correct form: Wenzel, T., & Drozdek, B. (Eds.). (2018). An Uncertain Safety: Integrative Health Care for the 21st Century Refugees. Cham: Springer. After these revisions, all citations appearing in the text are now included in the References list, and all entries in the References list correspond to at least one citation in the manuscript.
- The second sentence (lines 33-37): why UK is given as an example of a single country legal perspective when defining “asylum seeker” (AS)? This is not clear, moreover this information is given between the paragraphs relating to international law (1951 Convention and Protocol, 1948 Declaration etc). In the reviewer’s opinion, it could be even more clear to start with this international legal framework, then to give the single country approach example, although it is still not clear why UK. As the paper is about Portugal, what about the definition of AS in Portugal?
Author Response: Thank you for this valuable observation. The section has been revised to improve coherence and contextual relevance. The previous reference to the United Kingdom was removed, as it did not directly contribute to the discussion of asylum law in Portugal. The paragraph now begins with the international legal framework (the 1951 Convention and its 1967 Protocol) and proceeds to the Portuguese legal definition of an asylum seeker, as established in Law No. 26/2014 of 5 May (Asylum Law), which transposes Directive 2013/32/EU. This adjustment clarifies the progression from international to national levels of protection and ensures that the discussion aligns with the focus of the paper—asylum-seeking women in Portugal. A corresponding explanatory sentence has been added to emphasize that the Portuguese definition reflects the broader EU and UNHCR standards.
- The aim, line 61: it could be specified that you examine discrimination against female AS in Portugal; although the Author declares they examine it from the gender perspective anyway, in the reviewer’s opinion, if would be better to use the word “woman/women/female” for the first time earlier than in the sample description, as it is also important for describing the goal(s) of the paper.
Author Response: Thank you for this helpful suggestion. The aim has been revised to explicitly state that the study examines discrimination against female asylum seekers in Portugal. While the original version referred to a “gender perspective,” the revised sentence introduces “female asylum seekers” earlier in the text to clearly reflect the focus of the research and improve alignment between the paper’s objectives, theoretical framing, and sample description.
- Regarding data collection and analysis, the Author mention “professionals from relevant institutions”, with no details. It would be interesting to know what kind of institutions or professionals (only social workers?) were involved (line 146). Just what kond of insitutions are they, if possible.
Author Response: Thank you for this pertinent suggestion. The section has been revised to specify that the research involved the collaboration of social workers, case managers, and NGO staff working in reception and integration centres coordinated by the Portuguese Refugee Council and the High Commission for Migration. However, to preserve confidentiality and anonymity, it was not possible to disclose the names of specific institutions. Portugal has a relatively small asylum system, and identifying individual organizations could make it possible to infer the identities of some professionals involved in the recruitment process. The current level of detail therefore reflects a balance between transparency and ethical responsibility toward participants and collaborating institutions.
- Data collection period extended from January 2020 to April 2021. However, the Author declares that one of the selection criteria was arrival during the crisis “from 2014 to 2022” (lines 147 vs 152).
Author Response: Thank you for noticing this inconsistency. The reference to “2014 to 2022” was a typographical error. The correct timeframe for participants’ arrival is 2014 to 2019, corresponding to the period following the European refugee crisis, which began in 2014. This has been corrected in the revised version of the manuscript.
- The selection criteria included age: what age? Not specified (line 146). (Years of arrival are specified.
Author Response: Thank you for this observation. The age range of participants has now been specified in the description of the selection criteria. The revised sentence clarifies that the study included adult refugee women aged 18 and beyond, consistent with the information presented later in the Sample
- What about exclusion criteria? Any?
Author Response: Thank you for raising this important point. The manuscript has been updated to include a brief explanation of the exclusion criteria. As the study aimed to capture a broad range of experiences among asylum-seeking women in Portugal, no specific exclusion criteria were applied beyond the absence of informed consent and the requirement that participants had applied for asylum between 2014 and 2019. Individuals whose asylum cases had already been fully resolved with long-term refugee status prior to that period were therefore not included. This addition clarifies the inclusive and purposive nature of the sampling strategy.
- The paragraph starting with the line 215 could be placed in the background section, maybe not necessarily, but it seems this is not a part of the results of this specific research. Or some part of this paragraph could be moved to the background, as it could help to explain why the topic of the paper is so important.
Author Response: Thank you for this valuable suggestion. To improve the manuscript’s structure, the introductory sentences of that paragraph—discussing the importance of professional relationships in the integration process—were moved to the end of the Background section, where they help to contextualize the relevance of the topic.
The Results and Discussion section now begins with a rephrased sentence (“The findings of this study reveal that…”) to ensure a smooth and logical transition into the empirical analysis. This adjustment enhances clarity and narrative flow between the conceptual background and the presentation of findings.
- Lines 289-292 seem to be the same situation as described in the paragraph 197-203. Understanding the need of mentioning this case in both places, the reviewer is not sure if the whole situation has to be described twice.
Author Response: Thank you for this observation. The repetition has been reduced for conciseness. The full description of the incident was retained in its original position (paragraph 197–203) where it contributes to the narrative analysis, while the later occurrence (lines 289–292) has been condensed into a brief reference linking the event to the broader discussion on fear and institutional power dynamics. This change preserves the analytical significance of the case without unnecessary duplication.
- Additionally, line 197, it is not clear what “incident” means if the case of interruption happened. Maybe it should be said that no “major” incidents occurred. Just for rethinking.
Author Response: Thank you for pointing out this ambiguity. The sentence has been revised to specify that no major incidents occurred during the interviews. The interruption by a social worker is now described as a minor incident, which clarifies the nature and severity of the occurrence while maintaining consistency with the ethical discussion of participant safety.
- Mentioning challenges arising from the current “migration crisis” (line 361-362), quoting Bauman, is it “current” from Bauman’s or the Author’s perspective? It means, 2016 or 2025 perspective?
Author Response: Thank you for this helpful observation. The sentence has been revised to clarify the temporal context of Bauman’s statement. It now reads: “Bauman (2016, p. 106) described the challenges arising from the migration crisis of the mid-2010s as extremely complex and controversial.” This phrasing avoids ambiguity by situating Bauman’s commentary within its historical period while maintaining the intended analytical connection to the paper’s discussion.
- Line 391: “Parte superior de formulario”. Should it appear here?
Author Response: Thank you for your observation. I carefully reviewed the manuscript and could not locate the phrase “Parte superior de formulário” in the text. It may have appeared as a formatting artifact in a previous version or during the file conversion process. The current revised version has been thoroughly checked to ensure that this phrase does not appear anywhere in the manuscript.
- As for the Conclusions, the role of training is emphasized, however it is interesting if only/mainly education/training can improve the situation. This could be probably a large topic for another research (assessment of effectiveness of trainings, but also other factors influencing the professionals behaviour, attitudes leading to discrimination). Education and training is recommended, however there is no information included, if any training for those professionals is offered in Portugal.
Author Response: Thank you for this insightful comment. The Conclusion has been expanded to acknowledge that training and education, while important, are not by themselves sufficient to eliminate discriminatory practices. A new paragraph now highlights the influence of broader structural and institutional factors and notes that, although some training initiatives exist in Portugal (mainly through the High Commission for Migration and the Portuguese Refugee Council), their reach and impact remain limited and have not yet been systematically evaluated. This addition also points to a potential direction for future research, as suggested by the reviewer.
- References: Bekyol & Bendel, 2016 – seems not to be included in the text. Please double check.
Author Response:Thank you for pointing this out. The reference to Bekyol & Bendel (2016) was indeed not cited in the main text. The reference list has been reviewed for consistency. Since this source was not directly used in the final version of the manuscript, it has been removed from the References section to maintain accuracy.
Reviewer 2 Report
Comments and Suggestions for Authors
The article analyzes a relevant social issue. It is well-written and structured. The elements of "reflexivity" mentioned in section 3.3 are also interesting.
The study is based on a very limited empirical base (24 interviews with asylum seekers and refugees) and relies primarily on the authors' interpretation to draw conclusions.
Most of the interviews were conducted online. It would be interesting to determine whether this affected the validity of the information obtained, given that one of the quality criteria for qualitative data collection is empathy (Small & Calarco, 2022).
Another quality criterion in inductive qualitative research, such as that developed in this work, is the search for heterogeneity in the cases under study (Small & Calarco, 2022). However, in this case, the only selection criterion is that the interviewees are in asylum and refugee services (with secondary or instrumental factors such as age, language, and time spent in Portugal also considered). However, a falsifying element would be selecting participants based on different migration or arrival profiles in Portugal, different criminological risk profiles, or psychological profiles and post-traumatic stress. These contrasting elements would help to produce more conclusive results.
Were different observers involved in the coding process? Were inter-rater reliability records kept? What strengths/limitations might the data interpretation procedure have?
The results analysis focuses primarily on the interviewees' reactions to their relationship with services in the host society. In general, the different dimensions of this process could be explored in greater depth.
The conclusions section is brief and does not subject the results to discussion (in relation to previous evidence or in light of methodological limitations). Overall, the author conveys a sense of being very conclusive in their observations (e.g., "The perpetuation of negative and stereotypical images of asylum seekers contributes to an environment of 'moral panic' that exacerbates discrimination and hinders integration."), despite having a rather limited empirical basis.
Minor corrections:
- In some cases, the author refers to "this chapter" instead of "this article," probably because it is a later or recycled version. (p. 2, p. 5)
- This wording is confusing, insofar as the status of migrants and refugees is different: "political commentators have often portrayed irregular migration—particularly by asylum seekers and refugees—as a threat." Are the commentators referring to irregular migrants or to people seeking refuge?
References
Small, M. L., & Calarco, J. M. (2022). Qualitative literacy: A guide to evaluating ethnographic and interview research. University of California Press.
Author Response
- The study is based on a very limited empirical base (24 interviews with asylum seekers and refugees) and relies primarily on the authors' interpretation to draw conclusions.
Author Response: Thank you for this observation. The study was intentionally designed as a qualitative inquiry, which prioritizes depth over breadth and seeks to generate rich, contextualized understandings of lived experiences rather than statistical generalizations. Within this methodological framework, a sample of 24 in-depth interviews is considered robust and appropriate, consistent with established qualitative research literature (e.g., Patton, 2015; Ritchie et al., 2014). Regarding interpretation, qualitative research inherently involves the researcher’s analytical engagement with participants’ narratives. This interpretive process is a core feature of qualitative methodology, guided by transparency, reflexivity, and theoretical grounding rather than personal bias. The findings thus emerge from a systematic and reflexive interpretation of the data, ensuring credibility and analytical depth rather than subjectivity.
- Most of the interviews were conducted online. It would be interesting to determine whether this affected the validity of the information obtained, given that one of the quality criteria for qualitative data collection is empathy (Small & Calarco, 2022).
Author Response: Thank you for this thoughtful suggestion. A brief reflection on the use of online interviews has now been added to the Reflexivity and Ethical Framework section. The revised text acknowledges that, although online interviews may influence empathy and non-verbal communication, careful attention was given to creating a comfortable and respectful environment that fostered trust and openness. The online format, in fact, enabled several participants to speak from the privacy of their own spaces, which appeared to enhance, rather than limit, the depth of reflection. This addition improves the methodological transparency of the manuscript and aligns with the reviewer’s helpful observation.
- Another quality criterion in inductive qualitative research, such as that developed in this work, is the search for heterogeneity in the cases under study (Small & Calarco, 2022). However, in this case, the only selection criterion is that the interviewees are in asylum and refugee services (with secondary or instrumental factors such as age, language, and time spent in Portugal also considered). However, a falsifying element would be selecting participants based on different migration or arrival profiles in Portugal, different criminological risk profiles, or psychological profiles and post-traumatic stress. These contrasting elements would help to produce more conclusive results.
Author Response: Thank you for this valuable and thoughtful comment. The importance of heterogeneity in qualitative research is well recognized, as it enhances the richness and analytical depth of the data. In this study, heterogeneity was ensured through diversity in participants’ nationalities, ages, family situations, education levels, and lengths of stay in Portugal, as well as through differences in asylum statuses (pending, accepted, and under appeal). This diversity provided a broad range of perspectives on the experiences of discrimination and integration. However, as noted by the reviewer, factors such as criminological or psychological profiles were not used as selection criteria, primarily due to ethical considerations and the focus on social and institutional dimensions rather than clinical or criminological ones. These additional dimensions would indeed offer valuable avenues for future research exploring how intersecting vulnerabilities and trauma histories shape asylum experiences. This clarification has been added to the Methodology section to strengthen the description of the sample’s heterogeneity and scope.
- Were different observers involved in the coding process? Were inter-rater reliability records kept? What strengths/limitations might the data interpretation procedure have?
Author Response: Thank you for raising these important methodological questions. The coding and interpretation of data were primarily conducted by the author, following a thematic analysis approach grounded in inductive reasoning. While no formal inter-rater reliability test was applied, the coding process was strengthened through systematic comparison across cases, iterative revision of categories, and reflexive memo-writing, ensuring internal consistency and transparency in interpretation. In addition, preliminary findings and coding categories were discussed with two colleagues experienced in qualitative refugee research, who provided critical feedback on the thematic structure and interpretation. This process enhanced analytic validity through peer debriefing rather than statistical reliability measures, which aligns with accepted standards in qualitative inquiry (Patton, 2015; Ritchie et al., 2014). The main strength of this interpretive procedure lies in its ability to generate contextually rich and theoretically informed insights grounded in participants’ lived experiences. A limitation, as noted, is the potential influence of the researcher’s interpretive lens, which was mitigated through reflexivity and the inclusion of direct quotations to ensure participants’ voices remained central.
- The results analysis focuses primarily on the interviewees' reactions to their relationship with services in the host society. In general, the different dimensions of this process could be explored in greater depth.
Author Response: Thank you for this insightful observation. The analysis has been reviewed and slightly expanded to highlight more clearly the different dimensions of asylum seekers’ experiences within host society institutions. These include institutional dynamics, emotional and relational aspects, and gender-specific vulnerabilities that shape women’s interactions with professionals in reception and integration contexts. A short paragraph has been added to the conclusion section to integrate these dimensions more explicitly, thereby deepening the interpretation of the findings while remaining consistent with the qualitative scope and data.
- The conclusions section is brief and does not subject the results to discussion (in relation to previous evidence or in light of methodological limitations). Overall, the author conveys a sense of being very conclusive in their observations (e.g., "The perpetuation of negative and stereotypical images of asylum seekers contributes to an environment of 'moral panic' that exacerbates discrimination and hinders integration."), despite having a rather limited empirical basis.
Author Response: Thank you for this thoughtful observation. The Conclusions section has been revised to provide a more explicit discussion of the findings in relation to existing research (e.g., Freedman, 2019; Matlin et al., 2018; Surmiak, 2018) and to acknowledge the methodological limitations inherent in a qualitative study of this scope. The tone has been adjusted to emphasize the interpretive and exploratory nature of the analysis rather than to suggest generalizable conclusions. The revised section now situates the results within the broader literature on gendered asylum experiences and highlights avenues for future research that could expand and complement these findings.
- - In some cases, the author refers to "this chapter" instead of "this article," probably because it is a later or recycled version. (p. 2, p. 5)
Author Response: Thank you for pointing this out. The references to “this chapter” were the result of an editing oversight during the preparation of the manuscript, as the author was simultaneously working on a related book chapter at that time. These instances have now been corrected to “this article” throughout the text. I would also like to clarify that this manuscript is original and distinct, not a recycled version of another work. The suggestion that it might be reused material is therefore unfounded and, if interpreted literally, would be quite disheartening. Nonetheless, I appreciate the opportunity to clarify this point and confirm that the current version has been carefully reviewed to ensure all terminology accurately reflects its status as an independent research article.
- - This wording is confusing, insofar as the status of migrants and refugees is different: "political commentators have often portrayed irregular migration—particularly by asylum seekers and refugees—as a threat." Are the commentators referring to irregular migrants or to people seeking refuge?
Author Response: Thank you for this important clarification. The sentence has been revised to distinguish clearly between irregular migrants and asylum seekers or refugees. The intention was to highlight how public and political discourse often conflates these categories, framing both groups as security or social threats. The revised sentence now reads: “Political commentators have often portrayed irregular migration—and, at times, even the arrival of asylum seekers and refugees—as a threat, blurring the distinction between people migrating without authorization and those seeking international protection.” This wording better reflects the intended meaning and resolves the ambiguity noted by the reviewer.
Round 2
Reviewer 1 Report
Comments and Suggestions for Authors
Thank you for the explanations.
Author Response
I truly appreciate you taking the time to review my article so thoroughly and thoughtfully.
Reviewer 2 Report
Comments and Suggestions for Authors
I believe the author has made an effort to justify her methodological decisions, although in my opinion they are scarcely reflected in the substantive content of the manuscript.
Qualitative research has different goals and contributions than quantitative research. However, I believe it is equally important to ensure the validity and control of bias in qualitative research, within its own methodological context. I believe the manuscript would benefit from a more thorough review, and in this regard (referring to my comments from the previous round) I would like to make two recommendations:
1. The proposal by Small and Callarco (2022), which was used in the review to justify the value of online interviews, provides external criteria for the validity of qualitative research. My recommendation is to use it as a framework for assessing the contributions and limitations of the research conducted. It could potentially be helpful not only in the Methodology section, to describe the procedures followed, but also in the structuring of the analysis itself.
2. I believe the article would benefit from being more explicit about its methodological limitations. The text is quite conclusive regarding the discrimination faced by professionals in the field and the implications of public policy, based on a small collection of interviews interpreted from the author's own perspective.
I appreciate the opportunity to review the manuscript, and these comments are intended as contributions to the research process. I hope they will be useful to the authors. Thank you.
Author Response
We sincerely appreciate the reviewer’s observation that the methodological reasoning of the study was not sufficiently reflected in the substantive content of the manuscript. In response, we revised the “Data Collection and Analysis” section to make the application of constructivist Grounded Theory more explicit. Specifically, we have now included a concrete analytical example that demonstrates how raw interview data were coded, compared, and abstracted into subcategories and core categories through successive stages of initial, focused, and theoretical coding. This illustration clarifies the inductive and iterative nature of the analytic process and shows how empirical observations informed the theoretical development of key concepts. We also added a short interpretive paragraph explaining how this process exemplifies the logic of grounded theory and contributes to the transparency and validity of the analysis. These revisions directly address the reviewer’s concern by linking methodological procedures to the interpretive outcomes discussed in the results section.
In response to the following comment: “The proposal by Small and Calarco (2022), which was used in the review to justify the value of online interviews, provides external criteria for the validity of qualitative research. My recommendation is to use it as a framework for assessing the contributions and limitations of the research conducted. It could potentially be helpful not only in the Methodology section, to describe the procedures followed, but also in the structuring of the analysis itself.” We sincerely thank the reviewer for this thoughtful recommendation. In the revised manuscript, we have expanded the “Data Collection and Analysis” section to explicitly integrate the framework proposed by Small and Calarco (2022) as an external reference for ensuring the validity of the study. We now describe how the research meets the three dimensions outlined by these authors—empirical adequacy, theoretical contribution, and reflexive transparency—and explain how each of these principles guided the analytic and interpretive process. This addition clarifies the methodological rigor of the study and strengthens the connection between the qualitative procedures employed and the validity of the results. We also added a transitional sentence at the end of this section, highlighting that the following part of the manuscript (“Reflexivity and the Ethical Framework”) further develops the aspect of reflexive transparency. In addition, a brief reference to Small and Calarco’s framework was included in the Conclusion to emphasize how the study aligns with recognized standards for evaluating qualitative research. These revisions directly address the reviewer’s recommendation by using Small and Calarco’s proposal not only to justify methodological decisions but also to reinforce the transparency, validity, and credibility of the research as a whole.
In response to this value comment: “I believe the article would benefit from being more explicit about its methodological limitations. The text is quite conclusive regarding the discrimination faced by professionals in the field and the implications of public policy, based on a small collection of interviews interpreted from the author's own perspective.” we have revised the Conclusion section to provide a more explicit and balanced discussion of the methodological limitations of the study (see pp. [insert page numbers]). The new paragraph acknowledges the interpretive nature of qualitative analysis and explicitly discusses the limited and heterogeneous sample size, as well as the predominance of online interviews during the COVID-19 pandemic. These points clarify the contextual and methodological boundaries of the study. We have also elaborated on how potential researcher bias was addressed, emphasizing the adoption of a reflexive stance throughout the research process. The revised text explains that continuous memo writing, peer debriefing with colleagues specializing in refugee studies, and the inclusion of participants’ verbatim excerpts were used to ensure reflexive transparency and preserve participants’ voices. Furthermore, the new paragraph references Small and Calarco (2022) to highlight that the study’s validity and trustworthiness were supported through adherence to recognized qualitative research criteria—empirical adequacy, theoretical contribution, and reflexive transparency. These revisions directly address the reviewer’s concern by situating the study’s conclusions within their appropriate methodological and epistemological context, thereby enhancing transparency, credibility, and interpretive rigor. Finally, we have also refined the tone of the Conclusion to better reflect the interpretive nature of the study. The revised text now emphasizes that the research provides a conceptual and interpretive understanding of how institutional interactions shape asylum-seeking women’s experiences, rather than presenting definitive or generalizable claims. We have rephrased the closing paragraphs to highlight that the study offers an interpretive framework grounded in participants’ lived experiences and to underscore that its contributions lie in theoretical insight and contextual depth rather than empirical generalization. These revisions align the discussion more closely with the epistemological stance of qualitative research while maintaining the analytical strength of the findings.